# Synthetic Material Abdominal Swabs Reduce Activation of Platelets and Leukocytes Compared to Cotton Materials

**DOI:** 10.3390/biom11071023

**Published:** 2021-07-13

**Authors:** Katharina Gerling, Lisa Maria Herrmann, Christoph Salewski, Melanie Wolf, Pia Müllerbader, Dorothea Siegel-Axel, Hans Peter Wendel, Christian Schlensak, Meltem Avci-Adali, Sandra Stoppelkamp

**Affiliations:** 1Clinical Research Laboratory, Department of Thoracic and Cardiovascular Surgery, University Hospital Tübingen, Eberhard Karls University of Tübingen, 72076 Tübingen, Germany; Katharina.Gerling@uni-tuebingen.de (K.G.); lisa-maria.herrmann@student.uni-tuebingen.de (L.M.H.); Christoph.salewski@med.uni-tuebingen.de (C.S.); melanie.wolf@klinikum.uni-tuebingen.de (M.W.); Pia.Muellerbader@klinikum.uni-tuebingen.de (P.M.); Hans-Peter.Wendel@med.uni-tuebingen.de (H.P.W.); Christian.schlensak@med.uni-tuebingen.de (C.S.); meltem.avci-adali@uni-tuebingen.de (M.A.-A.); 2German Center for Diabetes Research (DZD e.V.) and Institute for Diabetes Research and Metabolic Diseases of the Helmholtz Center Munich at the Eberhard Karls University of Tübingen, 85764 Neuherberg, Germany; dorothea.axel@med.uni-tuebingen.de; 3Department of Internal Medicine IV, Division of Endocrinology, Diabetology and Nephrology, University Hospital Tübingen, 72076 Tübingen, Germany

**Keywords:** cardiopulmonary bypass surgery, systemic inflammation, heart–lung machine, cotton, abdominal swabs

## Abstract

During surgical procedures, cotton abdominal swabs with their high absorptive capacity and malleability are used to retain organs and absorb blood or other body fluids. Such properties of the natural material cotton are advantageous for most operations, but in cardiopulmonary bypass (CPB) surgery, a high blood volume can accumulate in the thoracic cavity that is quickly retransfused via the heart–lung machine (HLM). This common practice is supposed to be safe due to the high anticoagulation. However, in vitro analyses showed that blood cells and plasma proteins were activated despite a high anticoagulation, which can propagate especially an inflammatory response in the patient. Thus, we investigated patients’ blood during CPB surgery for inflammatory and coagulation-associated activation after contact to the HLM and either cotton or synthetic abdominal swabs. Contact with cotton significantly increased thrombocyte and neutrophil activation measured as β-thromboglobulin and PMN-elastase secretion, respectively, compared to synthetic abdominal swabs. Both inflammatory cytokines, interleukin (IL) 1β and IL6, were also significantly increased in the cotton over the synthetic patient group, while SDF-1α was significantly lower in the synthetic group. Our data show for the first time that cotton materials can activate platelets and leukocytes despite a high anticoagulation and that this activation is lower with synthetic materials. This additional activation due to the material on top of the activation exerted by the tissue contact that blood is exposed to during CPB surgery can propagate further reactions in patients after surgery, which poses a risk for this already vulnerable patient group.

## 1. Introduction

Abdominal swabs are often used in surgical procedures to absorb blood or body fluids from the surgical field or to retain organs [1]. Mostly cotton abdominal swabs are used due to the high absorbency; however, cotton as a natural product is subject to various influences during the growing process, which can affect the quality of the raw material [2]. In addition, the manufacturing process can influence blood compatibility [3]. Although contact time is usually only a few hours, blood components (cells and proteins) can get activated.

In most surgeries, where heavy bleeding is expected, such as hip replacement, open-heart, or aorta surgeries, blood is aspirated using a vacuum blood sucker and retransfusion of the aspirated blood is routinely performed to avoid allogeneic blood transfusions [4]. The coagulation, fibrinolysis, and inflammation in the sucked blood are strongly activated [5,6]. Furthermore, cells are exposed to shear stress, which can lead to their destruction [7]. Thus, the sucked blood is processed in a cell saver to remove cell fragments, human plasma, clotting factors, free hemoglobin, and tissue debris. Solely erythrocytes are retransfused to the patient [8], which prevents the transfer of activated mediators into the systemic circulation and thereby reduces complications [9,10].

Abdominal swabs are also routinely used in the field of cardiovascular surgery [11]. During cardiopulmonary bypass operations, the blood is pumped through the HLM and it comes into contact with foreign surfaces, resulting in the activation of cells and the secretion of pro-inflammatory and -coagulant mediators [12]. Thus, to prevent the activation of blood coagulation, the patient’s blood is anticoagulated with approximately 3 IU high-molecular-weight (HMW) heparin/mL blood, which leads to an activated clotting time (ACT) of >300–480 s. Since the blood is highly anticoagulated, in many heart centers, it is common that the blood from the surgical site is wrung out of the used abdominal swabs and sucked into the cardiotomy reservoir of the HLM. Then, the aspirated blood is retransfused without the purification via a cell saver, since a possible activation of the coagulation system for thromboembolic events is considered negligible [13].

In previous studies, a possible procoagulant potential of cotton abdominal swabs was detected using a simple clotting test [14]. Hypercoagulant swabs reduced the ACT and the concentration of free heparin [3]. Further studies showed that even sterilized abdominal swabs were able to activate inflammatory processes in human blood [15], which indicates that the material itself has a certain inflammation activation potential. The retransfusion of sucked blood, which came in contact with the cotton abdominal swabs, without purification might lead to a higher risk of postoperative complications, such as thromboembolic or inflammatory events.

For some years, there have been comparable abdominal swabs available made out of synthetic fibers e.g., viscose or polyester [16]. Most steps during the manufacturing process of synthetic fibers can be controlled, and therefore, it is possible to influence the quality and characteristics of the synthetic product. The previous observation that cotton swabs could induce an inflammatory response despite a high anticoagulation led us to the question of whether there might be a benefit of using a synthetic material with lower activating potential, as activated cells and cytokines retransfused could have an effect on the development of a systemic inflammatory response syndrome (SIRS). Therefore, we set out to investigate this in a small clinical study with 20 cardiac patients undergoing elective surgery involving the HLM, which is divided in two groups of 10 patients each. In one of the groups, cotton abdominal swabs were used in the thoracic cavity and in the other group, synthetic non-woven fiber swabs were employed. To analyze the effect of the two materials in cardiac surgery, blood counts, activation of coagulation, or inflammation with cytokine and chemokine secretion between groups were directly compared.

## 2. Materials and Methods

### 2.1. Abdominal Swab Materials

Cotton (white) abdominal swabs from different manufacturers were used, depending on the current status of the central stock at the University Hospital Tübingen (e.g., from Lohmann & Rauscher GmbH & Co. KG, Neuwied, Germany; Paul Hartmann AG, Heidenheim, Germany). The synthetic non-woven (white) abdominal swabs were obtained from Mölnlycke Health Care GmbH, Düsseldorf, Germany.

### 2.2. Preparation of Sample Swabs for in Vitro Tests

Commercially available processed abdominal swabs (Table 1) were prepared in a laminar flow clean bench with pyrogen-free materials. First, the abdominal swabs were cut into 2.5 × 1 cm pieces. For the following monocyte activation test (MAT), untreated cotton or synthetic abdominal swabs and pyrogen-impregnated abdominal swabs were prepared. The pyrogen solution (Lipopolysaccharide, LPS; 2000 endotoxin units (EU)/mL; World Health Organization, Geneva, Switzerland) of different concentrations (1 EU/mL, 10 EU/mL) was dried overnight on the material under pyrogen-free conditions, yielding a concentration of 0.1 and 1 EU/mL LPS in the test tubes, respectively, when incubated with blood.

### 2.3. Blood Sampling for in Vitro Tests

Whole blood was collected by venipuncture (Safety-Multifly^®^ 20 Gx3/4 TW needle; Sarstedt, Nümbrecht, Germany) in heparin- (19 I.U./mL, Sarstedt, Germany) or EDTA- containing monovettes (EDTA-K 1.6 mg/mL, Sarstedt, Germany) after donors were informed and gave their written informed consent. The blood of four healthy donors was pooled for each procedure. The blood pooling was carried out according to specific instructions in the European Pharmacopoiea (EP) [17]. The strict exclusion criteria for blood donors of the EP also applied: no non-steroidal anti-inflammatory drugs (e.g., ibuprofen) at least 48 h before blood sampling and no steroidal anti-inflammatory drugs 7 days prior to blood sampling. The blood collection procedures were approved by the research and ethics unit of the University of Tübingen (project approval number 287/2020BO2).

### 2.4. Monocyte Activation Test (MAT)

The MAT was performed with human whole blood as described before [15]. Briefly, the cotton or synthetic swabs (untreated or pyrogen-impregnated) were placed in sterile 2 mL DNA LoBind Eppendorf reaction tubes, and 1.5 mL of 1:10 diluted pooled human whole blood was added and incubated for 18 h at 37 °C. Increasing 2-fold concentrations from 0.0125 to 0.5 EU/mL LPS (concentrations correspond to the volume in the incubation to enable a comparison to a solid swab material) were added to the diluted blood to detect the immune activation potential of the pooled blood to pyrogens (standard dilution row). A concentration of LPS in the middle of the used standard curve (0.1 EU/mL) served as liquid LPS spike and as the lower concentration for LPS impregnation of the swabs. Half of the sterile, non-impregnated swabs were spiked with liquid LPS directly at the time of incubation (yielding 0.1 EU/mL in the incubation) to detect potential interferences with the swabs. The LPS-impregnated samples (0.1 EU/mL and 1.0 EU/mL) were incubated with blood under the same conditions as the sterile swabs.

After the incubation, the samples were centrifuged at 300 g for 5 min, and the supernatant was analyzed for cytokine secretion by enzyme-linked immunosorbent assay (ELISA). Cytokine production triggered by the abdominal swabs was measured by ELISA (human IL1β Duo-Set; R&D Systems/BioTechne, Wiesbaden, Germany) according to the manufacturer’s instructions. The quantification of secreted IL1β was performed using Gen5 software (Biotek, Bad Friedrichshall, Germany).

### 2.5. Patients for the Clinical Study

The study performed was reviewed and approved by the local institutional ethics committee of the Eberhard Karls University of Tübingen (project approval number: 301/2018BO2). The design of the study aims to examine the influence of the abdominal swab material on biomarkers of inflammation and coagulation activation. Therefore, 20 patients requiring cardiac surgery with the heart–lung machine (HLM) were enrolled and gave their written, informed consent. The patients were divided randomly into two groups. In group one, cotton abdominal swabs were used as is currently the standard procedure. In group two, a synthetic abdominal swab was used instead of cotton in the thoracic cavity during the whole cardiopulmonary bypass. Patient blood was collected at three time points: (1) from HLM directly after connection, (2) from HLM shortly before hemostasis, (3) wrung out from the abdominal swab after antagonizing heparin with protamine. The latter samples were obtained at the time of hemostasis from blood that would not be retransfused to safe patient blood during surgery. Although those samples were not exactly the same samples that were retransfused to the patients, they give an indication of the activating potential of the swab types. Only patients undergoing planned surgery with HLM support were recruited, and special attention was paid to only include patients that did not take immunosuppressant drugs. The two groups consisted of patients of similar age, weight, BMI, comorbidities, and hospitalization periods. The operations were performed under the same conditions. The duration of the operation, the bypass, and the aortic clamping were identical in both groups (Table 2). Additionally, inflammatory markers (C-reactive protein (CRP), procalcitonin (PCT), SIRS, and sepsis-related organ failure assessment (SOFA score) were evaluated from patient files. Criteria for SIRS were two or more of the following: temperature >38 °C or <36 °C, heart rate >90 beats/min, respiratory rate >20 breaths/min or paCO2 <32 mmHg, white blood cell count of >12.000 cells/mm^3^ or <4000 cells/mm^3^ [18]. The SOFA score was assessed as the sum of the severity scale (0–4) of six organ parameters: oxygenation index (pO_2_/FiO_2_) for lung, Glasgow Coma Scale for nervous system, mean arterial blood pressure for cardiovascular system, bilirubin for liver, thrombocytes for coagulation, and creatinine for kidney [19].

### 2.6. Plasma and Blood Sampling from Patients

During surgery, 3 different blood samples were taken: (1) After the establishment of the HLM connection, blood was taken directly from the HLM. (2) Towards the end of the surgery, during the phase of hemostasis, blood was again taken directly from the HLM. (3) Blood was collected from the blood-soaked abdominal swab by wringing out. Blood cell counts were additionally measured at patient admission, post-surgery, and at discharge. The blood was collected into ethylenediaminetetraacetic acid (EDTA; Sarstedt, Nümbrecht, Germany) containing monovettes for the measurement of the blood cell counts, complement activation ELISA, and multiplex immunoassay. CTAD (a mixture of citrate, theophylline, adenosine, and dipyridamol; BD Biosciences, Heidelberg, Germany) vacutainer were used for platelet activation ELISA, and citrate monovettes (Sarstedt, Nümbrecht, Germany) was used for coagulation activation (thrombin–antithrombin III complex; TAT) and leukocyte activation ELISAs (PMN-elastase). After the collection, the blood was analyzed or processed as soon as possible but after one hour at the latest. The blood cell counts (thrombocytes, erythrocytes, and lymphocytes) were measured directly with an automatic cell counter (ABX Micros 60, Horiba Medical, Kyoto, Japan). For all other activation markers, plasma was obtained by centrifugation (1800× *g*, 18 min at room temperature for citrated blood and 2500× *g*, 20 min at 4 °C for EDTA and CTAD blood). The plasma was shock frozen and stored at −20 °C (citrate and CTAD plasma) or −80 °C (EDTA plasma).

### 2.7. Soluble Activation Markers

The influence of the two different abdominal swab materials on the activation of the complement system, blood clotting, activation of platelets, and activation of neutrophils was evaluated by the detection of different activation markers using ELISAs, which were performed according to the manufacturer’s instructions. For this purpose, the shock frozen plasmas were used. The following activation markers were measured: terminal complement complex (sC5b-9) (MicroVue™ Complement, Quidel, Osteomedical GmbH, Sissach, Switzerland) in EDTA plasma, thrombin–antithrombin III complex (TAT) (Enzygnost^®^ TAT micro, Siemens Healthcare, Erlangen, Germany) and the polymorphonuclear (PMN)-elastase (PMN-elastase ELISA, demeditec, Kiel, Germany) in citrated plasma, β-thromboglobulin (β-TG) (Asserachrom^®^ β-TG, Diagnostica Stago, Düsseldorf, Germany) in CTAD plasma.

### 2.8. Cytokine and Chemokine Assessment

Cytokine concentrations of IL1β, IL6, SDF-1α, MCP-1, and TNFα were determined using a multiplex immunoassay (Human High Sensitivity Cytokine Luminex Performance Assay; R&D Systems/Bio-Techne, Wiesbaden, Germany) according to the manufacturer’s instructions. For the measurement, frozen EDTA plasma samples were diluted 1:4 with calibration diluent RD6-40 and measured with polystyrene microparticle beads in a Bioplex 200 (BioRad, Feldkirchen, Germany).

### 2.9. Statistics

The statistical analyses were performed using the software package GraphPad Prism version 6.01 (GraphPad Software Inc., La Jolla, CA, USA). After analyzing the samples for normal distribution by Shapiro–Wilk normality test, either Student’s unpaired *t*-tests or Mann–Whitney U tests were performed. Statistical significance was defined as *p* < 0.05.

## 3. Results

### 3.1. Preliminary In Vitro Test Shows Marked Differences between Materials

In a preliminary in vitro test, the inflammatory activation of blood by synthetic and cotton abdominal swabs was investigated using the monocyte activation test (MAT). As observed previously [15], cotton abdominal swabs themselves induced relatively strong secretion of the inflammatory cytokine IL1β, whereas blood incubated with the synthetic material showed no measurable IL1β secretion, actually the same response as blood without LPS or materials (Figure 1). The significantly different responses between swab types (*p* < 0.0001) indicate that the synthetic material does not activate the blood monocytes and thus shows different material properties than cotton, which induced a similar IL1β secretion in blood as the middle of the LPS standard curve (0.1 EU/mL) (Figure 1). On the other hand, the addition of liquid LPS to the incubating swabs did only marginally increase the cytokine secretion with a recovery rate of roughly 42% and 22%, respectively (Table 3). Two concentrations of LPS dried on the surface were almost not detectable any longer. According to the EP, LPS recovery rates below 50% show interference of LPS with the test substance [17]. Therefore, both materials, synthetic and cotton, inhibited the recovery of LPS, giving them similar properties in this respect. This behavior is generally advantageous for the abdominal swab materials, since this indicates that potential contaminants, e.g., when used for wounds, would be rather trapped than stimulate an immune response. Due to the interesting marked differences in the inflammatory response of whole blood to the materials alone, an in vivo study with two patient groups was planned and conducted.

### 3.2. Leukocyte Counts Increased at the End of HLM Only in the Cotton Group

As a first indicator of differences between operations performed with the two materials, blood cell counts were ascertained (Figure 2). In addition to the blood samples obtained during the operation, the blood cell counts from the patient’s records were used to get a more complete picture of the clinical outcome. No significant changes in platelet counts (Figure 2a) or red blood cell counts (RBC) (Figure 2b) were observed between cotton and synthetic abdominal swab groups, but significant decreases compared to admission values were seen within both groups. While RBC counts were similarly decreased compared to admission in both groups, platelet counts showed significant decreases in the cotton group for HLM start, HLM end, and post-surgery, but only for HLM start in the synthetic group. However, this was likely due to the generally larger variability of platelet counts in the patient group with synthetic swabs. The lower RBC counts during HLM is explained by the blood loss and dilution of blood with the priming fluid. The platelet and RBC counts in the blood samples obtained from the cotton and synthetic abdominal swabs are reduced and clearly in the pathological range, indicating a possible adhesion and activation of thrombocytes to the material and also trapping of RBCs in the swabs or hemolysis of RBCs.

Interestingly, there was only a significant increase in white blood cells (WBC) between HLM start and post-surgery in the synthetic group, whereas in the cotton group, the time points HLM end and post-surgery were significantly increased compared to admission and HLM start (Figure 2c). Moreover, the WBC count at the end of HLM was significantly elevated compared to discharge. Between groups, at the start of the HLM, WBC counts were significantly lower in the synthetic group than in the cotton group, but for both groups, the values were in the physiological range. Shortly before stopping the HLM, the WBC counts of patients from the synthetic group were still in the physiological range, but those from the cotton group were not. Additionally, leukocyte counts were significantly elevated in the cotton group at the end of HLM compared to the synthetic group. WBC counts in the blood obtained by wringing of the swabs were not significantly different between the two groups.

### 3.3. Hemocompatibility Parameters Differed for Leukocyte and Thrombocyte Activation

The activation of the complement system (sC5b-9), inflammation (PMN-elastase), coagulation (TAT), and platelets (β-TG) was analyzed in the blood samples from the beginning and end of HLM as well as in the blood samples obtained by wringing out of abdominal swabs (Figure 3). At the end of HLM, all measured hemocompatibility markers were significantly increased in both cotton and synthetic groups (Figure 3a–d), indicating that the activation of blood through the HLM treatment predominated over possible differences between the materials. While complement activation was similar between materials (Figure 3a), blood obtained from cotton swabs showed significantly higher activation of inflammation (PMN-elastase, Figure 3b), the coagulation cascade (TAT, Figure 3c), and platelets (β-TG, Figure 3d). The most marked difference was observed for PMN-elastase levels, where the activation of the blood by the synthetic material was only slightly higher than the activation seen at the end of HLM (Figure 3b). In contrast, cotton increased PMN-elastase levels to a much larger extend.

### 3.4. IL6 and IL1β Secretion was Lower with Synthetic Swabs

Similar to the in vitro experiments, inflammatory cytokine secretion was also investigated in vivo. Compared to the start of HLM, IL6 and TNFα levels were significantly increased in both groups at the end of HLM showing an inflammatory activation during HLM, while IL1β secretion was not significantly different (Figure 4). Between the groups, both IL1β and IL6 secretion was significantly lower at the end of HLM in the synthetic compared to the cotton group (Figure 4a,b), while TNFα secretion did not differ (Figure 4c). This difference between groups points toward an influence of the materials. Especially for IL6, this was also seen in blood obtained directly from the swabs (Figure 4b).

### 3.5. Levels of Chemokines MCP-1 and SDF1-α Reflect Inflammatory Processes

In addition to inflammatory cytokines, the chemokines monocyte chemoattractant protein-1 (MCP-1/CCL2) and stromal cell-derived factor 1 (SDF-1α) were measured. In our experiments, we could see an increase in MCP-1 levels during HLM (start HLM vs. end HLM) (Figure 5a), but not between groups or materials. SDF-1α was significantly increased at the end of HLM in both groups (Figure 5b). Interestingly, in blood wrung from the swabs, the synthetic material induced a significantly stronger activation of this marker than cotton.

## 4. Discussion

The aim of the current study was to ascertain whether or not abdominal swab material may influence blood parameters in addition to cardiopulmonary bypass surgery and thus reveal a possible source for unwanted additional activation of blood that could be avoided. The obtained data have to be discussed in two different aspects: (1) changes occurring during HLM/surgery reflect combined effects of the surgery, the HLM circuit, and the abdominal swab material, (2) parameters measured from blood that was wrung out of the abdominal swabs are mainly due to effects of the material and applied shear stress. As the surgery parameters and HLM contact times were not significantly different between groups, the differences observed at the HLM end between groups likely reflect the additional influence of the used swab material.

The significantly lower RBC and platelet counts at the beginning of HLM are explained by dilution of the blood with priming fluid [20]. The activation of blood parameters during cardiopulmonary bypass using the HLM has been described for the complement system, coagulation, leukocyte activation, and inflammatory cytokines [21,22,23]. The effects observed in our study reflect these observations to a large extend. Moreover, the similarities in blood cell counts between the blood samples wrung from the abdominal swabs indicates that the adhesion of platelets and leukocytes or the destruction of erythrocytes due to the shear stress from wringing [24,25] was reasonably similar between the materials.

However, the different activation parameters during HLM between patient groups point toward an additional effect of the material. Here, the samples taken from the swabs can be used to support the material-induced activation. The pathologically increased leukocytes at the end of HLM and post-surgery that were only observed in the cotton group cannot only be explained by the procedure but are likely due to stronger activation and cell proliferation [26] caused by the material. There are two possibilities of how the material-induced activation may have occurred: (1) directly on the leukocytes or (2) through triggering of other blood cells or proteins that affected the leukocytes. The natural material cotton consists of subunits belonging to the β-glucans, which are known to trigger an immune response in mammals [27] and plants [28]. Although it is not known whether this interaction of glucan molecules with cell surface receptors also applies to the material woven from cotton fibers [16], NF-kb activation over Toll-like receptor (TLR) 2-stimulation was observed in a monocyte cell line [29]. The strong IL1β secretion after contact of blood with cotton seen in our in vitro experiment strengthens the assumption that cotton stimulates monocytes to secrete IL1β. This cytokine, released by monocytes upon stimulation and inflammasome processing [30], is a very potent trigger of fever [31,32] but also stimulates endothelial cells to secrete NO and express adhesion molecules and chemokines, thus facilitating neutrophil extravasation [31,33,34]. However, thrombocytes have been reported to also liberate IL1β upon stimulation [35]. There is a strong mutual interaction and activation between platelets and leukocytes (especially neutrophils and macrophages) [36]. Therefore, the increase in leukocyte numbers only in the cotton group during HLM is likely related to the stimulation of proliferation and differentiation of neutrophil precursors by IL1β [30], which is released by either monocytes or platelets and can induce the mobilization of granulocyte progenitor cells and maturation of neutrophils in the bone marrow [37].

PMN-elastase, released from activated neutrophils, was significantly higher at the end of HLM in both groups, and in blood from the cotton swabs compared to the synthetic swabs. This suggests that in addition to the HLM procedure, the material itself induced increased neutrophil degranulation. Since the groups were not significantly different at the end of HLM, the activating effect induced by the extracorporeal procedure likely concealed the effect of the material. Therefore, the increase in neutrophil counts at the end of HLM observed only for the cotton group was not synonymous with neutrophil activation and degranulation alone, as PMN-elastase was similarly increased at the end of HLM in both groups. Neutrophil degranulation can be induced by several stimuli such as contact phase FXIIa, kallikrein [38], LPS [39], or IL8 [40]. The latter is produced by activated monocytes [41]. Activated platelets secrete thrombocyte activating (e.g., platelet-activating factor, ADP, von Willebrand Factor, [42]) but also neutrophil-activating (neutrophil-activating peptide 2) and other inflammation-triggering mediators such as reactive oxygen species [36]. Therefore, the significantly higher activation of platelets, measured by β-TG release, in blood wrung from cotton could be responsible for the increased neutrophil activation. Platelets themselves are readily activated at negatively charged surfaces [39] or by shear stress [43] from the HLM or wringing of the swabs. Tissue-factor expressing cells, such as activated monocytes [44], or neutrophil extracellular traps (NET) from activated neutrophils [45], can trigger the coagulation cascade [46], but also contact phase activation (by hydrolysis of FXII [47]) through the artificial surfaces of the HLM or swabs can occur and promote coagulation and inflammation [48,49,50,51]. A significant coagulation activation (measured by TAT complex formation) was observed in blood taken from the HLM despite the high anticoagulation regime used to prevent this [52,53]. In comparison to the levels during HLM, the very high TAT formation in blood taken from both swabs reflects in part the beginning protamine treatment antagonizing heparin and the adsorption of the anticoagulant heparin to the material [3], thus preventing thrombin formation less efficiently than during HLM. However, there was a significantly stronger activation in cotton than in synthetic swabs, pointing to a stronger activation of the coagulation cascade by stronger contact phase activation or platelet activation.

The significantly higher IL6 secretion in blood wrung from cotton swabs likely reflects a material-induced inflammatory activation of monocytes [15], whereas the significant increase at the end of HLM in the cotton compared to the synthetic group for both inflammatory cytokines (IL1β and IL6) can be induced by activated monocytes, platelets, or endothelial cells [54,55]. The SDF-1α expression can be upregulated by inflammatory mediators such as IL1β or TNFα [56,57]. In our study, this chemokine was significantly increased during HLM in both groups, but in blood wrung from swabs, the synthetic swabs induced a significantly higher expression than cotton swabs. This observation seems contradicting to the other observations as it has aggravating effects on platelet aggregation upon binding to the platelet surface receptors CXRC4 or CXCR7 [58]. In line with cotton’s stronger activation of platelets and leukocytes, we would therefore expect to see the opposite effect. Although it has long been assumed that SDF-1α originates mainly from bone marrow and endothelium [59], there is also a pool of SDF-1α released from platelets [60]. Especially this SDF-1α has been shown to promote platelet survival [58] and regulate monocyte differentiation and survival [61]. Thus, SDF-1α measured from swabs and during HLM may come from different sources. While SDF-1α levels in blood taken from HLM are not different between patient groups, the levels measured from swabs are, indicating a possible role of platelet-derived SDF-1α after direct contact with the materials and thus having a protective rather than a destructive role for platelets and monocytes in this case.

Our data strongly suggest that an inflammatory activation of monocytes, degranulation of neutrophils, and activation of platelets has occurred due to the material, not least due to the strong IL1β secretion in vitro in contact with cotton but the almost complete abrogation in contact with the synthetic material. Although most measured parameters were similarly elevated at the end of HLM in both patient groups, there was a significant increase in the inflammatory parameters at the end of HLM in the cotton group. This stronger activation of blood components by cotton than the synthetic material poses an additional risk for patients to develop a sterile inflammatory response (SIRS) when the activated blood components enter the systemic circulation. PMN-elastase has been reported to further cytokine release [62], and the other pro-inflammatory mediators and substances released from activated thrombocytes contribute as well. Looking at the CRP levels and SOFA score in both groups, we could not see a significant difference between groups. However, in the cotton group, half of the patients (five) showed symptoms of SIRS, whereas in the synthetic group, only two patients had SIRS symptoms. Therefore, the synthetic material seems to be the safer choice for patients in CPB surgery where blood is retransfused without using the cell saver to remove inflammatory mediators.

Similar observations concerning synthetic materials have been observed in other studies. Hernández et al., were able to show in patients with hemodialysis that the leukocyte activation was higher when cellulosic membranes were used than synthetic membranes [63]. In monkeys, oral sutures made from nylon showed nearly no inflammatory tissue reaction and therefore were superior to sutures made from cotton [64]. The same tendency of a lower inflammatory reaction to synthetic polymers than to cotton was visible in our study.

However, all operating surgeons in our study described the synthetic materials as inferior regarding absorbency and moldability. This subjective view may base on the surgeons being familiar with the cotton swabs. At least for the adsorptive capacity of the synthetic material polyurethane, there was no disadvantage over cotton materials seen even after the repeated use of one swab [16]. When comparing the spread of blood drops on this material, it was visible that the spread on the synthetic material was less. In clinical use, this could be an advantage in locating the exact origin of bleedings [16]. However, the described stiffness of the synthetic material is a disadvantage that needs to be considered. Modifications of synthetic materials to improve wettability or haptic properties may be applied such as the photografting of 2-hydroxyethyl methacrylate (HEMA), a possible material for a biocompatible hydrogel [65], onto polypropylene, which resulted in significantly increased absorbency and decreased water wetting time of the polymer [66]. Overall, patient safety in terms of adverse reactions such as SIRS and optimal properties during surgery need to be evaluated.

## 5. Conclusions

For the first time, the current clinical study provided evidence for an additional activation of blood components due to cotton abdominal swabs. Especially in cardiopulmonary bypass surgery, if no cell saver is used, the retransfusion of the activated blood components may pose a further risk for the already critically ill patients. Whether or not this necessarily contributes to serious complications cannot be predicted by this small study with limited patient numbers but requires a larger cohort comparing post-surgery complications in patients where the materials have been used. Our data unequivocally show that despite its advantageous properties, the natural material cotton activates blood components. Ideally, the cotton material would be modified to show less activation or the malleability of the synthetic material improved. Nevertheless, the common practice of retransfusion should be reconsidered when using cotton materials, as the increased cytokine secretion may contribute to or aggravate systemic inflammatory reactions following surgery. A cost–benefit analysis for the use of materials, cell saver, and allogenic blood products is advisable.

## Figures and Tables

**Figure 1 biomolecules-11-01023-f001:**
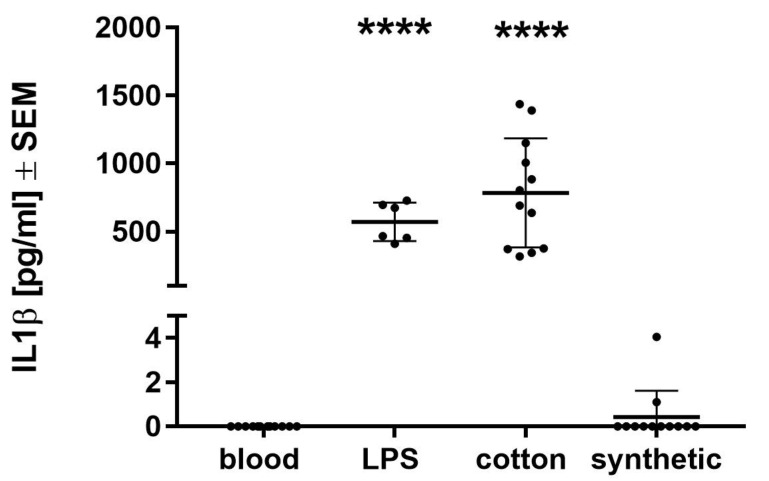
The MAT with human whole blood revealed inflammatory activation by cotton to a similar extend as 0.1 EU/mL liquid LPS added to blood, but there was no activation by synthetic swabs. Secretion of IL1β by blood monocytes is presented as scatter plot with means ± SEM. **** *p* < 0.0001, significantly different from blood without additives. One-way ANOVA with Tukey multiple comparison post hoc test.

**Figure 2 biomolecules-11-01023-f002:**
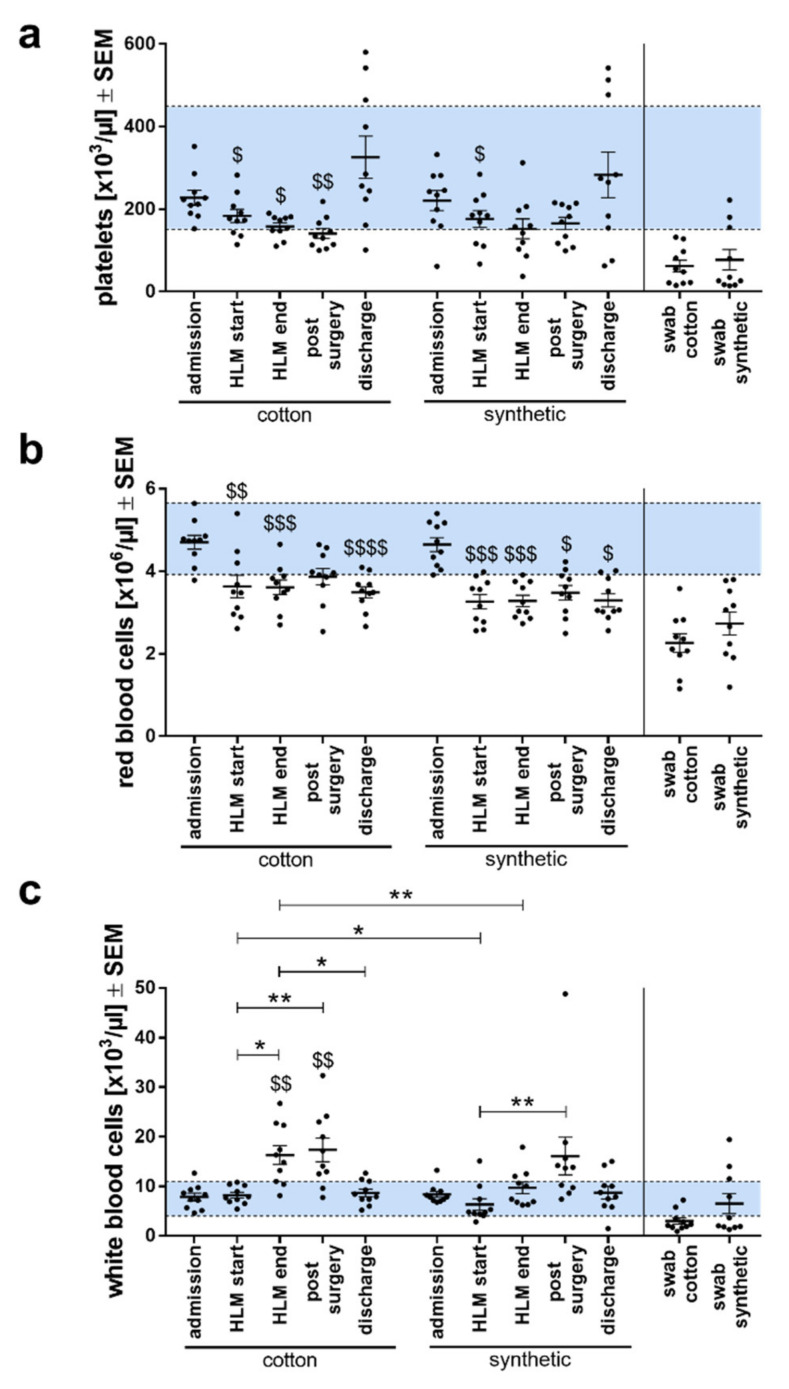
Analyses of blood cell counts of both patient groups (cotton swabs and synthetic swabs) measured at time of admission, during (HLM start, HLM end, and swab), and post-surgery as well as before discharge. (**a**) Platelets, (**b**) red blood cells, (**c**) white blood cells. Cell counts are presented as means ± SEM per study group. The horizontal line indicates the threshold values to pathological levels, the blue areas between the lines indicate the physiological ranges. Differences between groups were analyzed using Student’s *t*-test, differences within groups were analyzed using a one-way analysis of variance (ANOVA) with Bonferroni’s multiple comparison post-hoc test. The symbol “$” denotes significant differences compared to admission. * or $ *p* < 0.05; ** or $$ *p* < 0.01; $$$ *p* < 0.001; $$$$ *p* < 0.0001.

**Figure 3 biomolecules-11-01023-f003:**
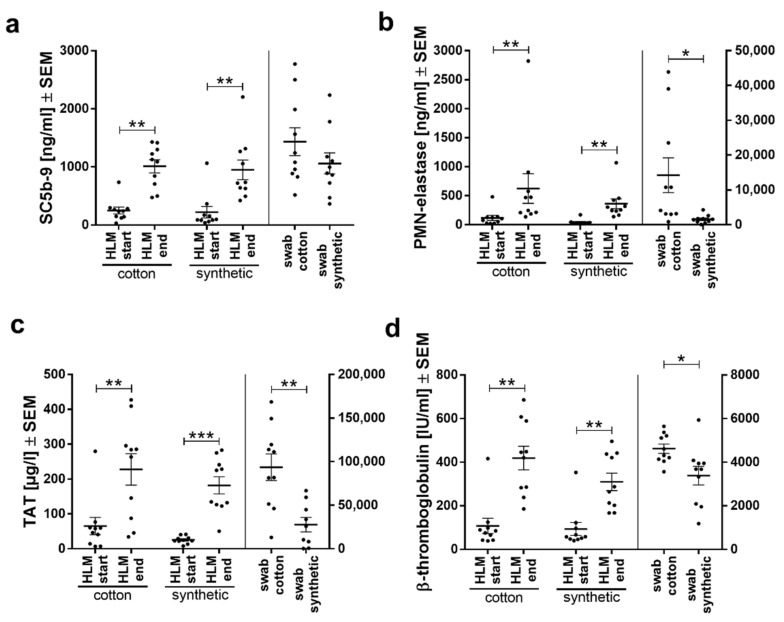
Hemocompatibility parameters measured from plasma of both patient groups indicate lower activation of neutrophils and platelets when synthetic swabs are used. Plasma levels of (**a**) complement, SC5b-9; (**b**) inflammation, PMN-elastase; (**c**) coagulation, TAT; and (**d**) platelet, β-TG activation from patients of both groups (cotton swabs and synthetic swabs) are presented as means ± SEM per study group. Differences were detected by Student’s *t*-test * *p* <0.05, ** *p* <0.01, *** *p* <0.001.

**Figure 4 biomolecules-11-01023-f004:**
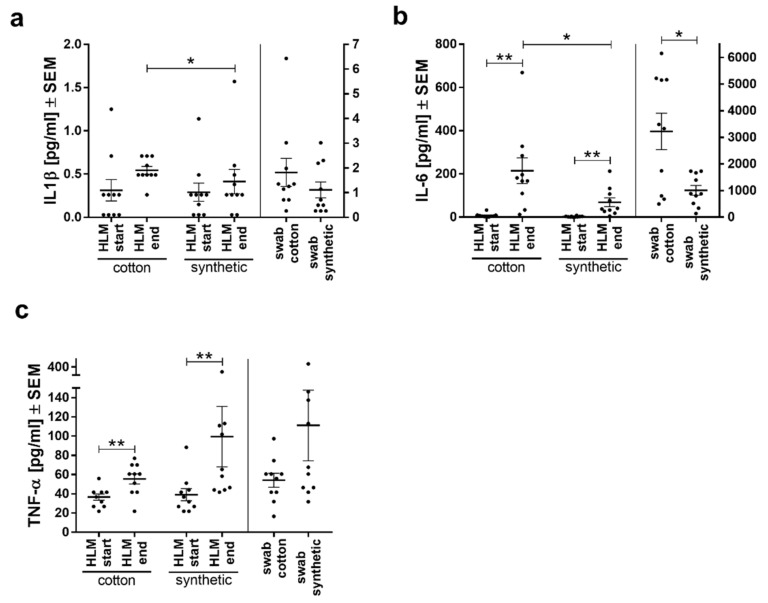
Interleukin levels of (**a**) IL1β and (**b**) IL6 from patients of both groups (cotton swabs and synthetic swabs) were lower with synthetic abdominal swabs than with cotton swabs, while no differences were seen with (**c**) TNFα secretion. Cytokine levels are presented as means ± SEM per study group. * *p* <0.05, ** *p* < 0.01, student’s *t*-test.

**Figure 5 biomolecules-11-01023-f005:**
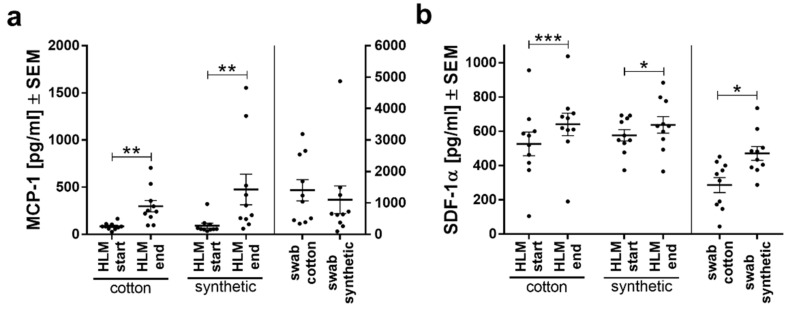
Plasma levels of chemokines are increased during HLM. Shown are plasma levels of (**a**) MCP-1 and (**b**) SDF-1α as means ± SEM per study group. Differences are measured with Student’s *t*-test. * *p* < 0.05, ** *p* < 0.01, *** *p* < 0.001.

**Table 1 biomolecules-11-01023-t001:** Overview of the abdominal swabs used to compare the pyrogenicity between cotton and synthetic abdominal swabs using the MAT.

Abdominal Swabs	Material	Lot	Company	Size
Telasorb white	cotton	299900005	Paul Hartmann AG	20 × 30 cm
BARRIER special non-woven abdominal swabs white	synthetic	17395462	Mölnlycke Health Care GmbH	40 × 40 cm

**Table 2 biomolecules-11-01023-t002:** Data of patients where cotton (*n* = 10) or synthetic (*n* = 10) swabs were used during the surgery. Data are presented as mean ± SD or percentage of patients per group. Differences between groups were assessed by Student’s *t*-test.

Patients	Cotton	Synthetic	*p* Values
Number	10	10 (1 death day 5)	
Age (years)	66.7 ± 7.0	68.9 ± 7.5	0.885
Gender (male/female)	7/3	6/4	0.639
Height (cm)	170.2 ± 10.1	169.3 ± 8.7	0.544
Weight (kg)	82.1 ±14.2	75.5 ± 20.1	0.244
BMI (kg/m²)	28.7 ± 4.7	25.9 ± 5.1	0.761
Body surface area (m^2^)	1.9 ± 0.2	1.9 0.3	0.272
Hospitalization (days)	13.4 ± 5.7	11.0 ± 3.5	0.118
Duration of surgery (min)	245.7 ± 47.0	229.0 ± 50.5	0.675
Duration of bypass (min)	133.7 ±45.7	103.6 ± 37.7	0.357
Duration of aortic clamping (min)	101.4 ± 36.1	76.3 ± 31.9	0.704
Intensive postoperative treatment (days)	2.0 ± 2.1	1.4 ± 1.2	0.305
Comorbidities			
BMI (kg/m²) <25	20%	30%	
BMI (kg/m²) ≥ 25-<30	50%	50%	
BMI (kg/m²) ≥30-<35	20%	20%	
BMI (kg/m²) ≥35-<40	10%		
Diabetes mellitus	20%	50%	
(Arterial) hypertension	60%	70%	
Inflammatory markers (post-surgery)			
Maximal CRP (mg/dl)	14.57 ± 5.88	12.16 ± 3.32	0.275
CRP at discharge (mg/dl)	4.26 ± 2.23	4.49 ± 2.67	0.855
Patients with PCT >2 (ng/mL)	2	1	
SIRS symptoms (1 day post-surgery)	5/10	2/10	
SOFA score (1 day post-surgery)	5.6 ± 3.5	4.2 ± 2.3	0.302

BMI: body mass index; CRP: C-reactive protein; PCT: procalcitonin; SIRS: systemic inflammatory response syndrome; SOFA score: sepsis-related organ failure assessment score.

**Table 3 biomolecules-11-01023-t003:** Proportional LPS recovery rates when added as liquid spike or when impregnated to the surface of the materials.

	Cotton	Synthetic
+0.1 EU/mL liquid LPS	42.3%	21.8%
+0.1 EU/mL impregnated LPS	0%	2.2%
+1.0 EU/mL impregnated LPS	0%	17.7%

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
