# Peer review of "Synthetic Material Abdominal Swabs Reduce Activation of Platelets and Leukocytes Compared to Cotton Materials"

_biomolecules, 2021, doi:10.3390/biom11071023_

Round 1

Reviewer 1 Report

The authors present a well-designed study of the effect of abdominal swab material on major coagulation and inflammatory markers. A major limitation of the study is the fact that the study does not include clinical outcomes for the patients studied, and as such the clinical relevance of this study is not able to be assessed. The study would be greatly improved by showing whether any of the differences in the laboratory-based parameters between the two swab types had any impact on the patient outcomes and therefore, whether the findings of this study have any clinical impact.

SPECIFIC MAJOR CONCERNS:

  1. The manuscript should include the actual p-values for all tables and figures where p<0.05 
  2. Table 2 should list the p-values for all inter-group comparisons.
  3. Line 206 to 220 seem to list instructions for authors and should not be a part of the manuscript.

MINOR COMMENTS:

  1. Please define HLM on line 56
  2. Please remove references to figures/tables from the discussion section.
  3. The authors should define the meaning of an "ordinary t-test". 

Reviewer 2 Report

The aim of the study is to compare the effect of cotton and synthetic swabs in cardiac surgery patients, on blood counts, activation of coagulation and inflammation. In vitro data are also provided. This is a interesting topic with a potential clinical relevance.

The paper is well written. The study well conducted. The results show significant differences between the two materials.

However, some revisions are needed.

Abstract. Thrombotic activation. This term is not correct. The coagulation parameter tested is the TAT complexes. The others are inflammation or platelet activation markers. All of these parameters reflect thrombin generation and inhibition, platelet activation, which may increase the risk of thrombosis.

Results: eventhough, the study was not designed and powered to evaluate the clinical outcome, it would be interesting to have few data on the outcome of the patients in the two groups (development of SIRS..)

The discussion should be revised.

- In their discussion, the authors make lot of assumptions on different mechanisms of inflammation activation that are not always supported by their results. Some parts of the discussion should therefore be shorten.

- Proposed mechanisms are mainly focused on monocyte activation while other cells can largely contribute to the inflammation state. Hemostasis and inflammation are intimately linked, and induce and amplify one another. Indeed, the interaction of platelets with leucocytes is a key step in both system activation which facilitates leukocyte extravasation to the site of inflammation and initiation of vascular inflammation (Mansour A et al, J Clin Med 2020). This issue should be addressed.

- In cardiac surgery, the sucked blood may be processed in a cell saver, which would remove cytokines, platelets and cell fragments. In this context, the differences between material may be less relevant. This should be considered in the discussion.

- Page 14, line 421 : the sentence “The strong increase of TAT…thus preventing thrombin formation less efficiently“. The increase of TAT in blood from taken from the swabs is more than likely due to activation of coagulation upon contact with swabs rather than an effect of protamine…

Reviewer 3 Report

The manuscript entitled “Synthetic material abdominal swabs reduce activation of platelets and leukocytes compared to cotton materials” compares differences in bio- and hemocompatibility of abdominal swabs made of cotton and synthetics based on a small clinical study. Of special interest is the reduced inflammatory response of the synthetic material both under in vitro and in vivo conditions.

Introduction:

The introduction is very long. Please shorten and get to the point.

Materials and Methods:

Materials and Methods are described very extensively and in detail. Very good!

What about the criteria for the patient selection? Randomization?

Table 2: P-values are missing! The subdivision of th BMI is incomplete!? The total should be 100% per group!? A BMI>=40 was not detected? Why is this subdivision mentioned? Replace the comma in „maximal CRP, cotton group“ with a dot (5,88).

Sampling of plasma/blood (section 2.6): A centrifugation of 25,000g (line 180) of blood resulted in plasma and destroyed cells!! Wrong number of revolution?

Statistics (Section 2.9): lines 206 -220 includes parts of the author guides. Please delete!

Results:

The results were presented in a clear and structured manner. However, the results are already being evaluated in the individual sections!? This should actually be part of the discussion. Is this a special requirement of the journal?

Part 3.1: Please remove lines 242-244 (author guides?). What is „EP“ (line 234)?

Figure 1: What about the baseline response of whole blood (no LPS, no cotton, no synthetic) – include these data into figure 1.

Part 3.2: Figure 2: Data from more than 3 points in time are listed (not mentioned in the Method section). Is there a time dependency in the individual groups? Especially when comparing „start vs end HLM“. Legend: The origin of the last two columns (swab connon and swab synthetic) in the figures is not apparent from the legend. Please insert (also in the following figures!). Ist he unit correct fort he RBCs (x106/myL)?

Page 12, Figure 4. The scaling of IL1beta and IL6 is poorly choosen so that the differences are not clear!

Discussion:

The discussion should be revised and restructured. The results are repeated. It would be important to discuss the effect of the HLM on the one hand (comparison of start vs. end of HLM) and the effect of the various materials (comparison of cotton vs. synthetics) on the other hand.

Summarizing the clinical and experimental data showed that almost all parameters were increased due to the HLM, that is a known phenomenon! Please include relevant references.

Of special interest is the reduced inflammatory response to the synthetics. These differences should be worked out more clearly, especially against the background that the HLM and the bypass surgery alone triggers an inflammatory response. Using the synthetic swabs could reduce additional activation.

All in all, it is a very clear and important study. The results could have an impact on the continued use of abdominal swabs in large surgical procedures.

Round 2

Reviewer 1 Report

The authors have adequately addressed the comments and have revised the manuscript accordingly.

Author Response

Thank you very much for assessing the changes to the manuscript.

Reviewer 3 Report

Congradulations. The changes have increased the quality of the mansucript!

I have only a few notes on Table 2:

The significance information belongs in a separate column entitled "p-values". The table should be given a brief title. The rest should be part of the text or the legend (that is missing). 

The BMI is missing the unit.

Author Response

Thank you very much for this comment, I have changed table 2 accordingly and the data are presented much clearer now.